# Tropical Peatland Forest Biomass Estimation Using Polarimetric Parameters Extracted from RadarSAT-2 Images

**Mirza Muhammad Waqar [1],\*** , **Rahmi Sukmawati [2]**, **Yaqi Ji [1]**
**and Josaphat Tetuko Sri Sumantyo [1]**

1   Center for Environmental Remote Sensing (CEReS), Chiba University, 1-33 Yayoi, Inage,
    Chiba 263-8522, Japan; jiyaqi0921@chiba-u.jp (Y.J.); jtetukoss@faculty.chiba-u.jp (J.T.S.S.)
2   Faculty of Mathematics and Natural Sciences, Padang State University, Jalan, Air Tawar Padang,
    Sumatera Barat 25111, Indonesia; rahmi.sukmawati27@gmail.com
\*   Correspondence: mirzawaqar@chiba-u.jp; Tel.: +81-90-6100-5446

**Abstract:** This paper was aimed at estimating the forest aboveground biomass (AGB) in the Central Kalimantan tropical peatland forest, Indonesia, using polarimetric parameters extracted from RadarSAT-2 images. Six consecutive acquisitions of RadarSAT-2 full polarimetric data were acquired and polarimetric parameters were extracted. The backscattering coefficient ($\sigma_o$) for HH, HV, VH, and VV channels was computed respectively. Entropy (H) and alpha ($\alpha$) were computed using eign decomposition. In order to understand the scattering behavior, Yamaguchi decomposition was performed to estimate surface scattering ($\gamma_{surf}$) and volume scattering ($\gamma_{vol}$) components. Similarly following polarimetric indices were computed; Biomass Index (BMI), Canopy Structure Index (CSI), Volume Scattering Index (VSI), Radar Vegetation Index (RVI) and Pedestal Height ($p_h$). The PolSAR parameters were evaluated in terms of their temporal consistency, inter-dependence, and suitability for forest aboveground biomass estimation across rainy and dry conditions. Regression analysis was performed between referenced biomass measurements and polarimetric parameters; VSI, H, RVI, $p_h$, and $\gamma_{vol}$ were found significantly correlated with AGB. Biomass estimation was carried out using significant models. Resultant models were validated using field-based AGB measurements. Validation results show a significant correlation between measured and referenced biomass measurements with temporal consistency over the acquisition time period.

**Keywords:** forest biomass; polarimetric parameters 2; radar vegetation index (RVI); volume scattering index (VSI); canopy structure index (CSI)

## 1. Introduction

Aboveground biomass is an important biophysical describing all living biomass above the soil that includes stems, branches, leaves, barks, seeds and foliage in terrestrial ecosystems [1]. It plays an important role in maintaining the carbon cycle by removing $CO_2$ from the atmosphere by the process of photosynthesis and storing it in the components of trees. Due to rapid urbanization, forest regions are deforested and degraded especially in developing countries. The tropical rainforests are the most significant carbon reservoir. They are home to gigantic trees, world-famous plants, birds and a variety of fascinating mammals. Around 80% of the world's documented species can be found in tropical rainforests, although they cover only 6% of the Earth's land surface. Furthermore, tropical rainforests have the largest living biomass and home to the highest rate of terrestrial biodiversity However they are the most endangered habitat and vulnerable to deforestation and degradation. Annual deforestation

rate of rainforests is alarmingly high which is about 140,000 km$^2$ [2]. Mostly rainforests are deforested by logging companies for timber and local community for farming. Among the most endangered rainforests are the south–east Asian rainforest especially in the Kalimantan Island is facing deforestation and degradation at an alarming rate especially due to a rapid increase in human population [3]. Due to deforestation, degradation and forest burning, the stored $CO_2$ can return back to the atmosphere and can alter atmospheric composition which can result in climate change and global warming [4].

Cost-effective assessment of the forest biomass is vital for effective forest industry, sustainable forest management and resource planning [5]. Traditional practices involve extensive fieldwork with substantial human resources in the field, however, such practices are not sustainable for developing countries like Indonesia. Recently remote sensing is being extensively used for forest studies [6–8], however, incorporating field-based forest biophysical parameters increases the estimation accuracy of forest biophysical parameters using remote sensing data [9–13]. Researchers attempted to estimate forest AGB using optical, SAR and lidar data. Each of these have the potential to estimate different characteristics of forest structure. Application of optical remote sensing is very limited to low stand biomass regions. However, high-resolution optical data provide estimation of biophysical parameters at stand level [14]. However, Synthetic Aperture Radar (SAR) and lidar are proven to be more effective over the medium to high stand level biomass [15]. Many researchers used optical remote sensing data ranging from medium resolutions to very high resolution that includes: Landsat, Sentinal, QuickBird and WorldView-2 for forest aboveground biomass estimation [16–20]. As tropical regions are mostly covered with clouds, the applications of optical remote sensing data is limited over the tropical area. However, SAR provides unique penetration capability through clouds, which allow all weather condition monitoring capability over tropical regions. Remote sensing of forest structure and biomass with SAR bear significant potential for mapping and understanding of ecological processes [21–23]. SAR can provide significant information about forest structure depending on microwave (X-, C-, L-, P-) band used for image acquisition. Scattering from X-band image mostly contain information about leaves and small branches, scattering from C-band image provide information about main branches, L-band have a penetration capability until stem of the tree, however, under ideal conditions P-band can penetrate until soil and main roots.

Because of it's unique penetration capability, SAR is very suitable for forest biophysical parameter estimation over a heterogeneous environment like natural forest. Polarimetric SAR (PolSAR) is also considered to be an alternative with active development particularly in forest applications. In this paper, PolSAR term will be used frequently for full polarimetric SAR or quad pol SAR data. PolSAR-based forest biophysical parameters estimation is an active research area nowadays where decomposition based polarimetric parameters are being used in synergy with field based reference measurements for forest biomass estimation [24–27]. PolSAR-based AGB estimation at higher level of forest biomass experiences saturation of PolSAR signal and it's well documented limitation of this technique [5,28,29]. As natural forest is perfectly random in nature, polarimetric parameters describing randomness of the target could be significantly correlated with forest biophysical parameters. Few attempts were made to exploit these polarimetric parameters to estimate forest parameters [30], however, a comprehensive analysis of polarimetric parameters and it's temporal consistency using time-series C-band PolSAR data is not performed yet. This study focuses on estimating aboveground biomass using polarimetric parameters estimated from C-band RadarSAT-2 image. Furthermore, consistency of these polarimetric parameters was investigated for its multi-temporal behaviour for AGB estimation.

The paper is organized as follows. Section 2 mainly focused on description of study area, PolSAR data used in this study, Ancillary Data used, referenced data used for modeling biomass as a function of polarimetric parameters, main PolSAR concepts with related description, preprocessing of PolSAR data and methodological framework adopted in this research. In Section 3, the key analysis was preformed and discussed. In Section 4, study is concluded and future directions are potentials are listed.

## 2. Materials and Method

### 2.1. Study Area

The study site is located in tropical peat and kerangas forests around Palangkaraya, the capital city of Central Kalimantan Province, Indonesia (Figure 1). Central Kalimantan lies within the Inter-Tropical Convergence Zone (ITCZ), and it falls under the wet tropical climate region. Central Kalimantan is hot and humid, the mean daily temperature ranges from 24 °C to 30 °C and annual rainfall varies between 2500 to 2800 mm [31–33].

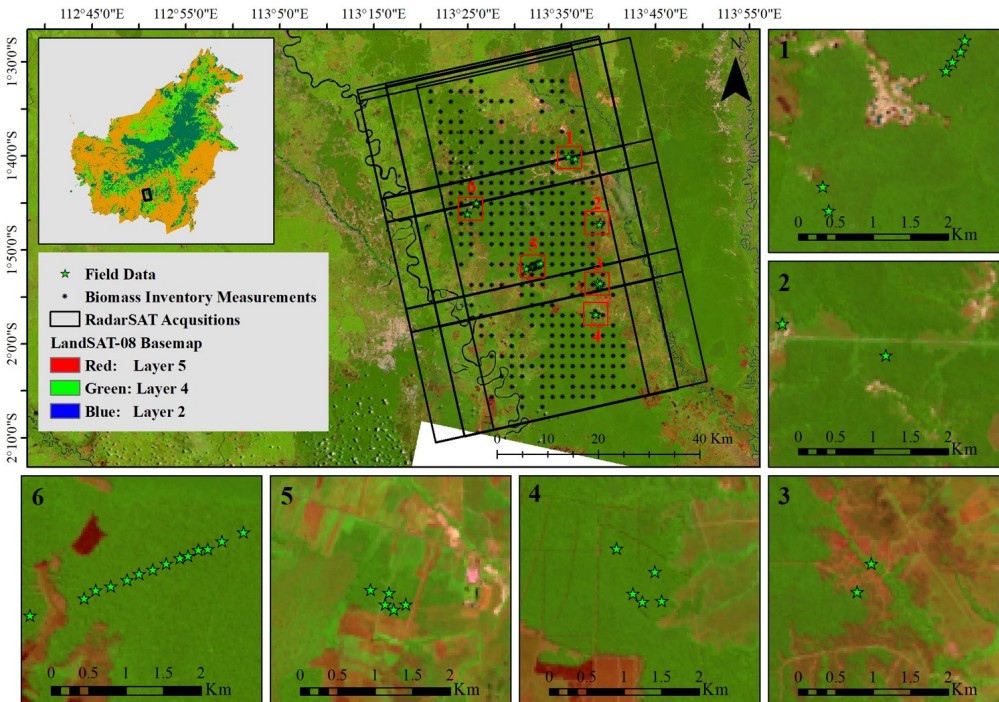

**Figure 1.** Location of the study area.

Rainfall in the study area is common throughout the year, however, the rainy season is from October to February and the dry season is from March to September. The forest canopy has three strata with a maximum height of 35 m. The principal tree species of the upper canopy are Gonystylus Bancanus, *Shorea* spp. (Meranti), Cratoxylon Glaucum (Gerongang) and Dactylocladus Stenostachys (Mentibu). The mix swamp forest grades into the low-pole forest, which continues for a further 7 km from the Sebangau river or so. Low canopy forest has only two strata and very few trees of commercial value. The principal species of the upper canopy are Combretocarpus Rotundatus (Tumeh), *Palaquium* sp., Dyera Costulata, Ilex Cymosa, *Dyospyros* sp. and *Calophyllum* spp. [34,35]. The study site is relatively flat with an elevation that varies between 4 m to 157 m. Rainfall is common throughout the year, and varies from about 60 inches (150 cm) to over 180 inches (450 cm) per year. In most parts of Sabah the wettest months occur during the North–East Monsoon from October through February and the driest months during the South–West Monsoon from March to September.

### 2.2. SAR and Ancillary Data

RadarSAT-2 full polarimetric times series data were acquired from the Canadian Space Agency (CSA), in single look complex (SLC) format. A total of six acquisitions were acquired for study site during October 2018 to January 2019 (see Figure 2 at incident angle ranging form 22° to 40° (see Table 1). Each acquisition of the study area comprises of three adjust scenes. Land cover of the study site was obtained from CIFOR Atlast for Borneo Island (https://www.cifor.org/map/atlas/). SRTM 30 m DEM of study site was acquired from EarthExplorer (https://earthexplorer.usgs.gov/).

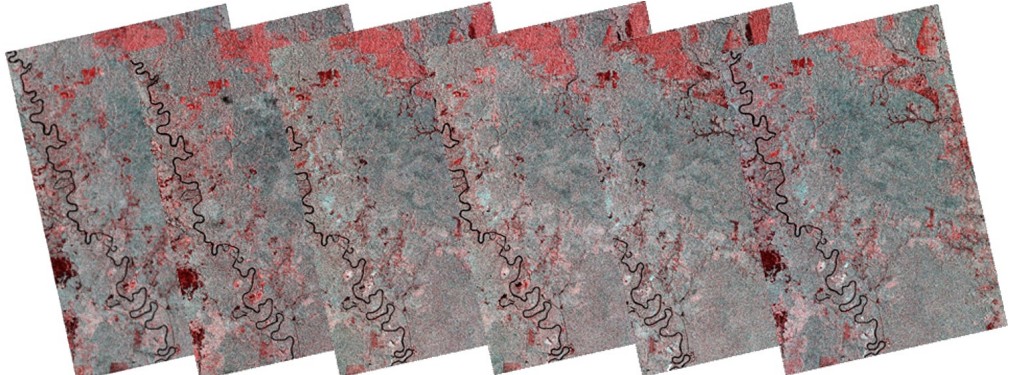

**Figure 2.** RadarSAT-2 time series acquired from October 2018 to January 2019.

**Table 1.** RadarSAT-2 scenes used in this study.

| Date of Acquisition | Acqusition Mode | Look Angle | Range and Azi Resolution |
|---|---|---|---|
| 20 October 2018 | Fine Quad Pol | 22.51~25.96 | 4.73 × 4.98 |
| 13 November 2018 | Fine Quad Pol | 22.51~25.96 | 4.73 × 4.98 |
| 24 December 2018 | Fine Quad Pol | 30.56~33.64 | 4.73 × 4.69 |
| 31 December 2018 | Fine Quad Pol | 22.49~25.96 | 4.73 × 4.98 |
| 10 January 2019 | Fine Quad Pol | 37.68~40.38 | 4.73 × 4.77 |
| 17 January 2019 | Fine Quad Pol | 30.56~33.64 | 4.73 × 4.69 |

Daily maximum temperature (°C) and precipitation measured from nearest meteorological weather station indicate the weather conditions during SAR data acquisition shown in Figure 3. The majority of the acquisitions were acquired under dry conditions, however, one acquisition was acquired under rainy conditions.

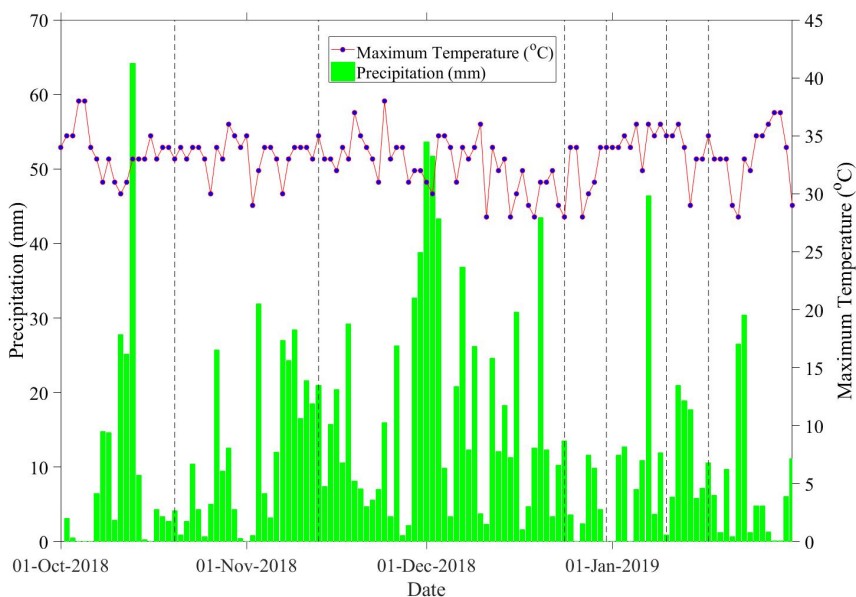

**Figure 3.** The daily precipitation and maximum temperature (from 1 October 2018 to 31 January 2019), the six vertical dash lines indicate SAR observation dates as listed in the Table 1.

## 2.3. Referenced Data

Referenced data were collected both from referenced biomass map [36] and a 30-day field visit in the study site. A total of 300 referenced measurements were taken from referenced map uniformly distributed throughout the study area and 54 plots of 20 m × 20 m dimensions were sampled during the field survey. Distribution of referenced biomass measurements taken from referenced biomass map and field plots are shown in Figure 1. In each field plot, the diameter at breast height (DBH), tree species, and plot center GPS location was measured. Due to the existence of wildlife in the study site and limited available resources, field data was only collected over easily accessible forest patches. A locally developed generic allometric equation [37] was used to calculate the stand level aboveground biomass. Table 2 listed locally developed generic allometric equations for the study site. The histograms for referenced AGB data are shown in Figure 4. Sample points collected from referenced biomass maps ranged from 0.04 Mg ha$^{-1}$ to 636 Mg ha$^{-1}$ and field data collected from field visit ranged from 28.36 Mg ha$^{-1}$ to 530.25 Mg ha$^{-1}$. Sampled data collected from referenced biomass map was used as training data and data collected from field survey was used as validation data.

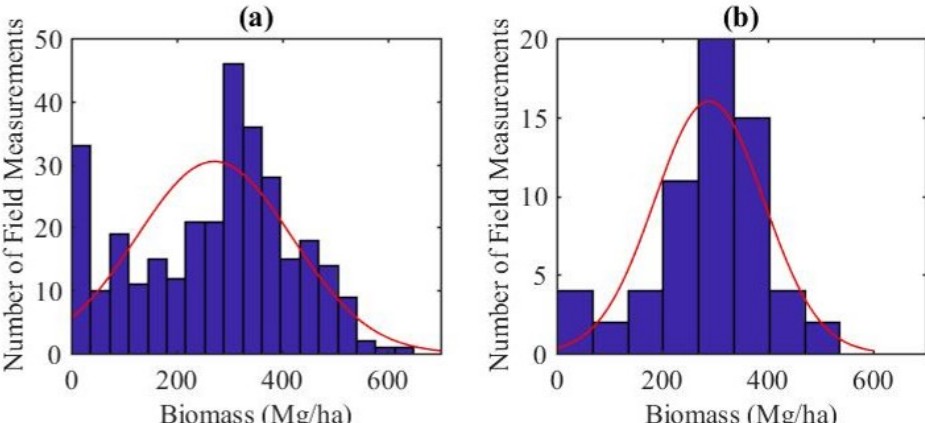

**Figure 4.** Forest aboveground biomass distribution. (**a**) Distribution in samples collected from referenced biomass map; (**b**) Distribution in data collected through field visit.

**Table 2.** Locally developed allometric equations for mix forest, where St, Br, Tw, Le and WSG are stem, branch, twing, leaf and wood specific gravity, respectively.

| Allometric Model | Sample | Tree Component | DBH (cm) | $R^2$ | Reference |
|---|---|---|---|---|---|
| lnAGB = −3.408+2.708 * lnD | 40 | St | 1.1–115 | 0.98 | [38] |
| AGB = 2.708 * D$^{2.486}$ | bda | St | 2–35 | 0.90 | [39] |
| lnAGB = −2.26 + 1.27 * lnD$^2$ | 184 | St | 4.8–69.7 | 0.99 | [40] |
| lnAGB = −4.26 + 1.36 * lnD$^2$ | 184 | Br + Tw | 4.8–69.7 | 0.91 | [40] |
| lnAGB = −3.86 + 1.01 * lnD$^2$ | 184 | Le | 4.8–69.7 | 0.81 | [40] |
| lnAGB = 1.201 + 2.196 * ln(D) | 122 | St | 6.5–200 | 0.96 | [37] |
| lnAGB = −0.744 + 2.188 * log(D)+0.832 * log(WSG) | 122 | St | 6.5–200 | 0.97 | [37] |
| lnAGB = −2.289 + 2.649 * ln(D)−0.021 * ln(D)$^2$ | 226 | St | 5–148 | 0.98 | [41] |
| AGB = 42.69−12.8(D) + 1.242(D$^2$) | 170 | St | 5–148 | 0.84 | [42] |

## 2.4. Main PolSAR Concepts in the Context of This Study

Fully polarimetric SAR measurements can be represented by scattering matrix shown in equation below:

$$[S] = \begin{bmatrix} S_{HH} & S_{HV} \\ S_{VH} & S_{VV} \end{bmatrix}, \tag{1}$$

where $S_{xy}$ is the complex backscattering term associated with $x$ and $y$ being the transmitted and received polarization respectively. Equation (1) can be rewritten in the Pauli basis:

$$\vec{k} = \frac{1}{\sqrt{2}} \begin{bmatrix} S_{HH} + S_{VV} & S_{HH} - S_{VV} & 2S_{HV} \end{bmatrix}^T. \tag{2}$$

In space-born SAR polarimetry, after polarimetric calibration, Faraday rotation compensation need to be applied by rotating it an angle $\theta$ around the radar line of sight leading to:

$$S' = \begin{bmatrix} cos\theta & sin\theta \\ -sin\theta & cos\theta \end{bmatrix} \begin{bmatrix} S_{HH} & S_{HV} \\ S_{VH} & S_{VV} \end{bmatrix} \begin{bmatrix} cos\theta & -sin\theta \\ sin\theta & cos\theta \end{bmatrix} \tag{3}$$

where

$$S' = \begin{bmatrix} S_{hh} & S_{hv} \\ S_{vh} & S_{vv} \end{bmatrix}. \tag{4}$$

The corresponding covariance matrix is positive semi-definite Hermitian:

$$[C_3] = \begin{bmatrix} \langle S_{hh}S_{hh}^* \rangle & \langle \sqrt{2}S_{hh}S_{hv}^* \rangle & \langle S_{hh}S_{hv}^* \rangle \\ \langle \sqrt{2}S_{hv}S_{hh}^* \rangle & \langle 2S_{hv}S_{hv}^* \rangle & \langle \sqrt{2}S_{hv}S_{vv}^* \rangle \\ \langle S_{vv}S_{hh}^* \rangle & \langle \sqrt{2}S_{vv}S_{hv}^* \rangle & \langle S_{vv}S_{vv}^* \rangle. \end{bmatrix} \tag{5}$$

The covariance matrix is fundamental to characterizing the SAR image to corresponding scattering components, e.g., surface, double-bounce, and volume scattering. Cloude and Pottier have proposed a polarimetric coherence matrix, reformulating the covariance matrix in the Pauli basis, with the target vector in the reciprocal mono-static case given by Equation (2). Then, the coherence matrix can be expressed as follows:

$$[T_3] = \frac{1}{2} \begin{bmatrix} \langle |S_{hh} + S_{vv}|^2 \rangle & \langle (S_{hh} + S_{vv})(S_{hh} - S_{vv})^* \rangle & \langle 2(S_{hh} + S_{vv})S_{hv}^* \rangle \\ \langle (S_{hh} - S_{vv})(S_{hh} + S_{vv})^* \rangle & \langle |S_{hh} - S_{vv}|^2 \rangle & \langle 2(S_{hh} - S_{vv})S_{hv}^* \rangle \\ \langle 2S_{hv}(S_{hh} + S_{vv})^* \rangle & \langle 2S_{hv}(S_{hh} - S_{vv})^* \rangle & \langle 4|S_{hv}|^2 \rangle \end{bmatrix} \tag{6}$$

2.4.1. Yamaguchi Decomposition Parameters

Yamaguchi proposed a four-component decomposition scheme [43], which can decompose a coherency matrix to the surface, double-bounce, volume and helix scattering. Mathematical expressions to compute volumetric scattering coefficients are listed below:

$$\gamma_v = 8 \langle |S_{HV}|^2 \rangle - 4 |Im \langle S_{HV}^*(S_{HH} - S_{VV}) \rangle|. \tag{7}$$

Corresponding volumetric scattering power can be obtained by:

$$P_v = \gamma_v. \tag{8}$$

Biomass Index (BMI) is an indicator of the relative amount of woody compared to leafy biomass. As BMI is not a ratio, and therefore is influenced by slope.

$$BiomassIndex(BMI) = \frac{\sigma_{HH}^o + \sigma_{VV}^o}{2} \tag{9}$$

Canopy Structure Index (CSI) is a measure of relative importance of vertical versus horizontal structure in the vegetation. Ecosystems dominated by nearly vertical trunks or stems will have higher CSI values than will ecosystems dominated by horizontal or near-horizontal branches. As the chosen study area for this research is intact forest, CSI can be an important indicator of tree density.

$$CanopyStructureIndex(CSI) = \frac{\sigma^o_{VV}}{\sigma^o_{VV} + \sigma^o_{HH}}. \tag{10}$$

Volume scattering index (VSI) is a measure of depolarization of the linearly polarized incident radar signal. High values of result when the cross-polarized bacscatter is dominating if compared to co-polarized backscatter.

$$VolumeScatteringIndex(VSI) = \frac{\sigma^o_{HV}}{\sigma^o_{HV} + BMI}. \tag{11}$$

The radar vegetation index (RVI) measures the randomness of scattering and can be written as:

$$RadarVegetationIndex(RVI) = \frac{8\sigma^o_{HV}}{\sigma^o_{HH} + \sigma^o_{VV} + 2\sigma^o_{HV}}. \tag{12}$$

Charbonnueau et al. (2005) assumed that $\sigma^o_{HH} \approx \sigma^o_{HH}$. This assumption is valid when the interaction between the soil and vegetation is negligible. Thus equation for RVI reduced to the form of:

$$RadarVegetationIndex(RVI) = \frac{4\sigma^o_{HV}}{\sigma^o_{HH} + \sigma^o_{HV}}. \tag{13}$$

Durden et al. (1990) [44] put forward a pedestal height ($h_p$) as the ratio of minimum eigenvalue to maximum eigenvalue (Lee and Pottier, 2009) [45]:

$$PedestalHeight(h_p) = \frac{min(\lambda_1, \lambda_2, \lambda_3))}{max(\lambda_1, \lambda_2, \lambda_3))}. \tag{14}$$

As a measure of the unpolarized backscattered energy, the pedestal height is expected to be high in the case of an forest area.

### 2.4.2. Eigen Decomposition Parameters

Cloude and Pottier (1996) proposed the following description for the eigenvectors of the covariance matrix in the Pauli basis [46]:

$$\tilde{e} = \begin{bmatrix} cos\alpha & sin\alpha cos\beta e^{i\delta} & sin\alpha sin\beta e^{i\gamma} \end{bmatrix}. \tag{15}$$

The average angle $\alpha$ can be calculated using

$$Alpha(\alpha) = P_1\lambda_1 + P_2\lambda_2 + P_3\lambda_3, \tag{16}$$

where

$$\lambda_1 = \frac{1}{2}|S_{HH} + S_{VV}|^2 \tag{17}$$

$$\lambda_2 = \frac{1}{4}|S_{HH} - S_{VV}|^2 + |S_{HV}|^2 + ImS^*_{HV}(|S_{HH} - S_{VV}|^2) \tag{18}$$

$$\lambda_3 = \frac{1}{4}|S_{HH} - S_{VV}|^2 + |S_{HV}|^2 - ImS^*_{HV}(|S_{HH} - S_{VV}|^2). \tag{19}$$

Entropy is the measure of target randomness or disorder, which is defined as:

$$Entropy(H) = -\sum_{i=1}^{3} p_i \cdot log_3(p_i); \qquad p_i = \frac{\lambda_i}{\sum_{k=1}^{3} \lambda_k}. \tag{20}$$

2.4.3. Backscattering Coefficient

The radar backscattering coefficient $\sigma^o$ provides information about the imaging surface, and it is the function of radar observation parameters: frequency, polarization, incident angle, and surface parameters: roughness, geometric shape and dielectric constant of the target.

$$\sigma^o_{slc} = 10 \cdot log_{10} < I^2 + Q^2 > + CF_1 + A. \tag{21}$$

*2.5. PolSAR Data Pre-Processing*

The acquired time series RadarSAT-2 full polarimetric data were multi-looked to 20 m square pixel. Backscattering coefficient ($\sigma_o$) for HH, HV, VH and VV channels were calculated using Equation (21). Eigen-decomposition parameters, i.e., alpha ($\alpha$) and entropy (H) were computed using Equations (16) and (20) respectively. Volumetric scattering behaviour was estimated using Yamaguchi decomposition (Equation (8)). Similarly polarimetric indices that include Canopy Structure Index (CSI), Volume Scattering Index (VSI), Radar Vegetation Index (RVI), Pedestal Height ($p_h$) were calculated using Equations (10), (11), (13) and (14) respectively. The whole processing was done using PCI Geomatica and open source SNAP software package provided by the European Space Agency (ESA). All of the computed parameters were terrain-corrected using SRTM 30 m DEM. Lee sigma filter $7 \times 7$ was performed to smooth the speckles in the resultant images. The same processing was applied on all RadarSAT-2 acquisitions as listed in Table 1. All of the polarimetric parameters extracted and used for further analysis are listed in Table 3.

**Table 3.** Polarimetric parameters used in this study.

| Polarimetric Parameter | | Description |
|---|---|---|
| Backscattering Coefficient | $\sigma_{HH}$ | Backscattering Coefficient of HH Channel |
| | $\sigma_{HV}$ | Backscattering Coefficient of HV Channel |
| | $\sigma_{VH}$ | Backscattering Coefficient of VH Channel |
| | $\sigma_{VV}$ | Backscattering Coefficient of VH Channel |
| Eighn Decomposition Parameters | H | Entropy |
| | $\alpha$ | Alpha |
| Yamaguchi Decomposition Parameters | $\gamma_{surf}$ | Surface Scattering |
| | $\gamma_{vol}$ | Volume Scattering |
| Polarimetric Parameters | CSI | Canoopy Structure Index |
| | VSI | Volume Scattering Index |
| | RVI | Radar Vegetation Index |
| | $p_h$ | Pedestal height |

Correlation analysis was performed among all polarimetric parameters listed in Table 3, and results are shown in Figure 5. As can be seen from the Figure 5 mostly the correlation among polarimetric parameters is not significantly high except few exceptions; e.g., $\alpha$ is correlated to $\sigma_{VH}$ with $R^2$ of 0.60; entropy (E) is significantly correlated with VSI, RVI and $p_h$ with $R^2$ of 0.60, 0.78 and 0.79 and $\sigma_{VH}$ is moderately correlated with VSI.

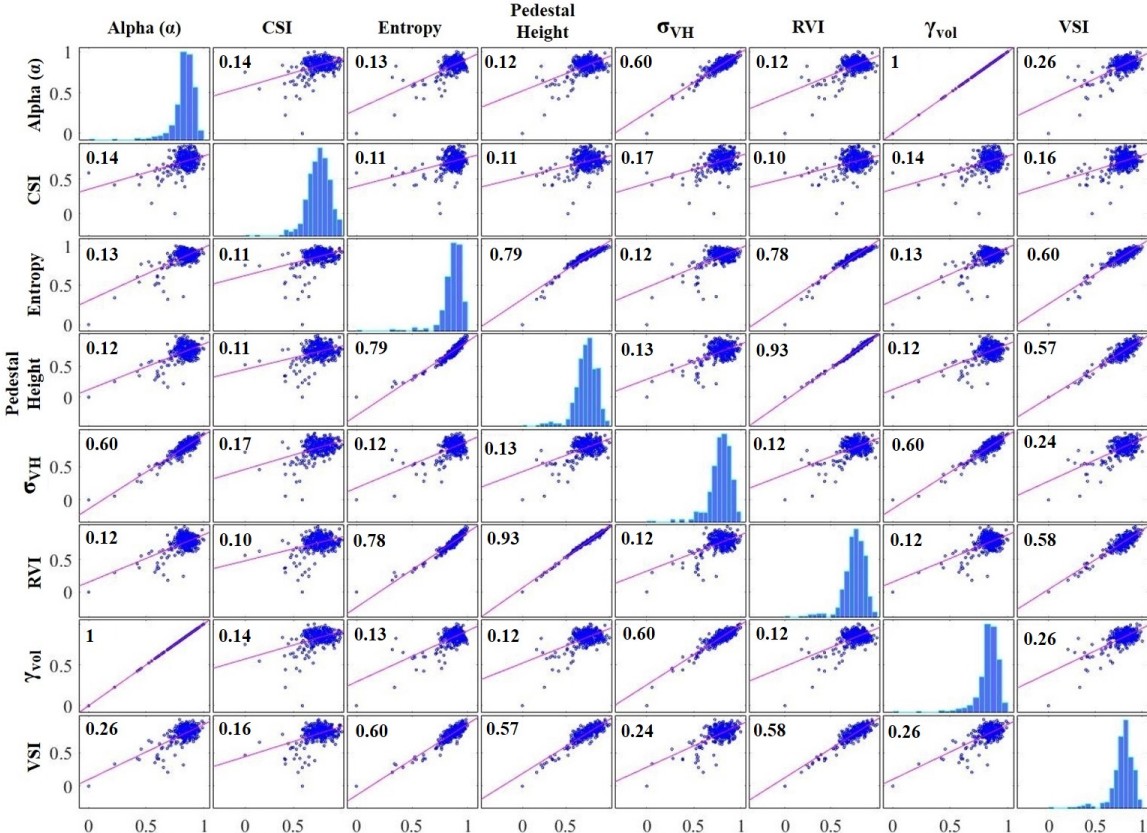

**Figure 5.** Correlation matrix among polarimetric parameters used in this research.

### 2.6. Modeling AGB vs. Polarimetric Parameters

As the referenced biomass measurements (training data) was made available at the 1 hectare scale. To match this, all polarimetric parameters were upscaled to 1 hectare (100 m × 100 m) resolution. Regression analysis was performed between referenced biomass measurements and polarimetric parameters. Logarithmic regression was chosen based on plotting results. In logarithmic regression saturation often occurs beyond a certain point. Hence, the accuracy such relationship significantly influenced beyond the saturation point. If $X$ is the independent variable and $Y$ is dependent variable, then the logarithmic regression equation can be written as:

$$Y = x_o + x_1 lnX \qquad (22)$$

where $X$ is the forest aboveground biomass in Mg ha$^{-1}$, $Y$ is the polarimetric parameter extracted from RadarSAT-2 image, $ln$ is the natural log and $x_o$, $x_1$ are regression coefficients. In case regression modeling, coefficient of determination ($R^2$) is of utter most significance. $R^2$ describes the proportion of variance in the dependent variable that is predictable from the independent variable.

$$R^2 = 1 - \frac{Unexplained Variation}{Total Variation} \qquad (23)$$

$$R^2 = 1 - \frac{SS_{res}}{SS_{tot}}. \qquad (24)$$

In the best case, if the relationship between dependent and independent variable is perfectly linear then $SS_{res} = 0$ and $R^2 = 1$.

Similarly, the resultant root mean square error can be estimated using equation written as under:

$$RMSE = \sqrt{\sum_{(i=1))}^{N} [\frac{(x_o - x_1)}{N}]}, \tag{25}$$

where $x_o - x_1$ is the residual and $N$ is the number of sample points.

The resultant regression analysis is discussed in Section 3. The same approach was adopted for all RadarSAT-2 acquisitions, regression results are summarized in Table 4. In order to get AGB as a function of polarimetric parameter, the resultant regression models were inverted. Using inverted regression biomass maps were generated for all acquisitions and using validation data, these maps were validated. The complete methodological framework is shown in Figure 6.

**Table 4.** Regression results for RadarSAT-2 acquisitions.

| Parameters | 20 October 2018 | 13 November 2018 | 24 December 2018 | 31 December 2018 | 10 January 2019 | 17 January 2019 |
|---|---|---|---|---|---|---|
| VSI | 0.61 | 0.45 | 0.51 | 0.64 | 0.60 | 0.55 |
| H | 0.58 | 0.43 | 0.49 | 0.60 | 0.57 | 0.53 |
| RVI | 0.53 | 0.43 | 0.49 | 0.57 | 0.55 | 0.51 |
| $p_h$ | 0.51 | 0.36 | 0.47 | 0.53 | 0.54 | 0.49 |
| $\gamma_{vol}$ | 0.44 | 0.32 | 0.49 | 0.49 | 0.46 | 0.43 |
| $\alpha$ | 0.44 | 0.33 | 0.41 | 0.48 | 0.46 | 0.44 |
| $\sigma_{VH}$ | 0.40 | 0.39 | 0.43 | 0.49 | 0.48 | 0.46 |
| CSI | 0.33 | 0.27 | 0.31 | 0.41 | 0.42 | 0.34 |

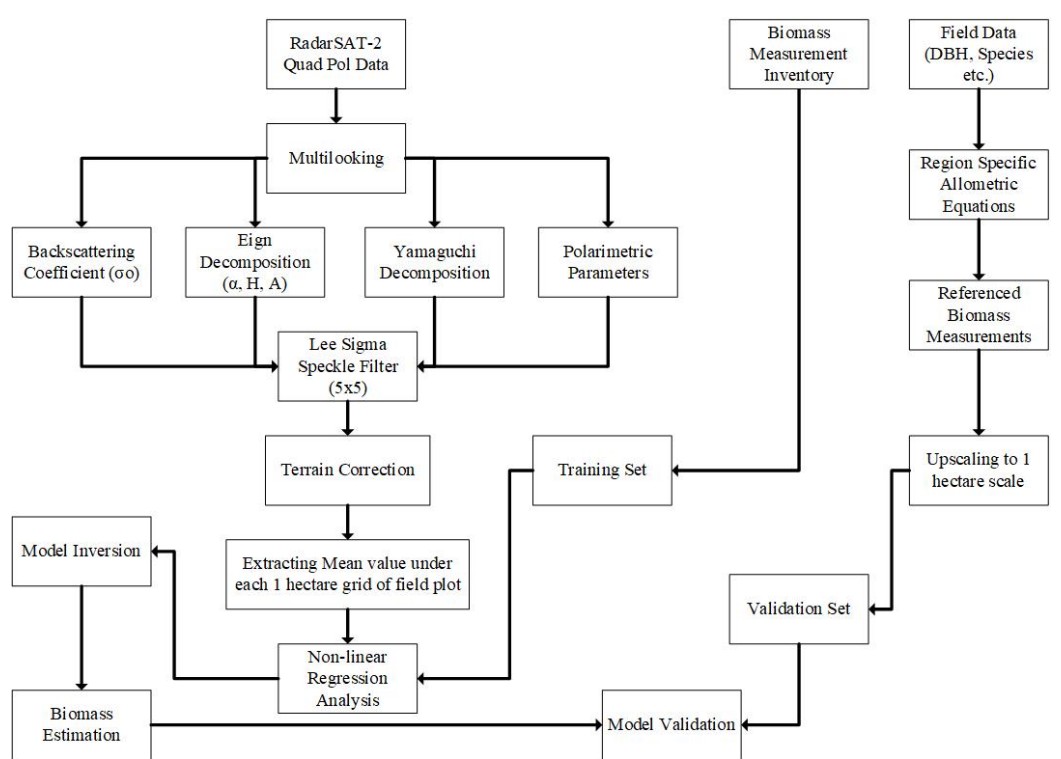

**Figure 6.** Methodological framework.

## 3. Results and Discussions

First of all, temporal consistency and inter-dependency of polarimetric parameters and their suitability for AGB estimation is discussed. We also described the results of regression between polarimetric parameters and reference biomass measurements. Next, the validation results are described in detail.

### 3.1. Temporal Dependence of Polarimetric Parameters

As PolSAR images were acquired under different weather conditions (see Figure 3). It is important analyze the impact caused by different weather conditions. For this, the PolSAR parameters extracted from temporal images were evaluated in terms of their temporal consistency, inter-dependence and suitability for forest aboveground biomass estimation across rainy and dry conditions. Figure 7 shows the correlation graph of each polarimetric parameter for all PolSAR acquisitions. It can be seen that $\alpha$ and CSI are not temporally correlated, however H, $\sigma_{VH}$, $p_h$, RVI, $\gamma_{vol}$ and VSI are temporally correlated at scale of moderate to high. Based on this temporal correlation, it is expected that the AGB vs polariemtric parameters modeling results will be temporally consistent except for $\alpha$ and CSI. The suitability of selected polarimetric parameters (listed in Table 3) for AGB estimation can be validated by modeling referenced AGB vs. polarimetric parameters using regression. For this, all selected polarimetric parameters (listed in Table 3) were chosen for further analysis.

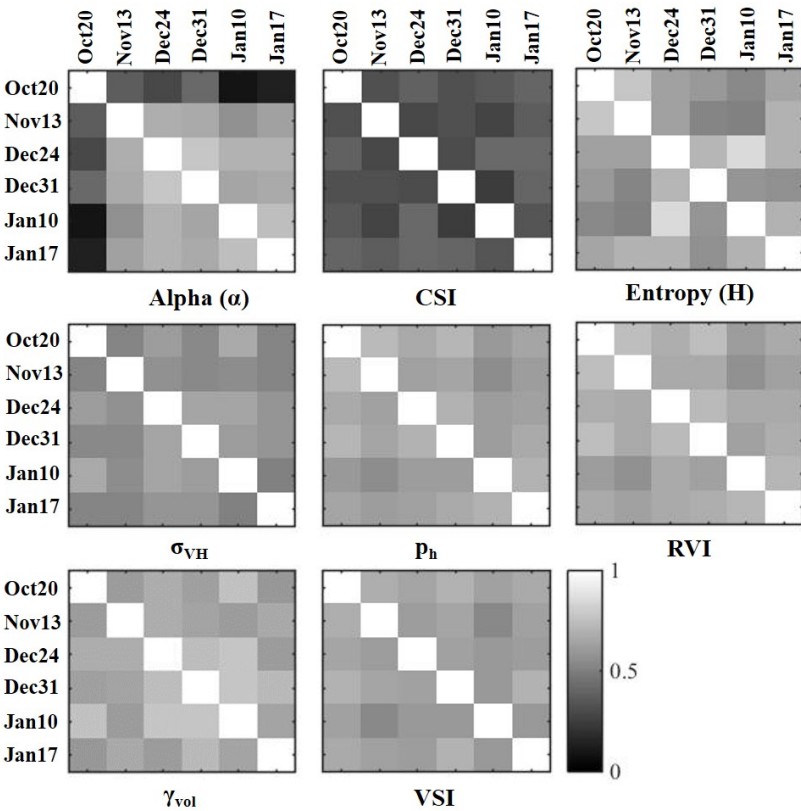

**Figure 7.** Correlation matrix illustrating the level of temporal correlation among polarimetric parameters extracted from RadarSAT-2 time series acquisitions.

### 3.2. Regression Analysis—Modeling AGB vs. Polarimetric Parameters

Regression results for AGB vs. polarimetric parameters for 31 December 2018 acquisition is shown in Figure 8. Similarly, regression results for selected polarimetric parameters for all PolSAR acquisitions are listed in Table 4. As it can be seen from the Figure 8, all polarimetric parameters were normalized between 0 and 1. This makes it easy to compare results from different polarimetric parameters having a different dynamic range.

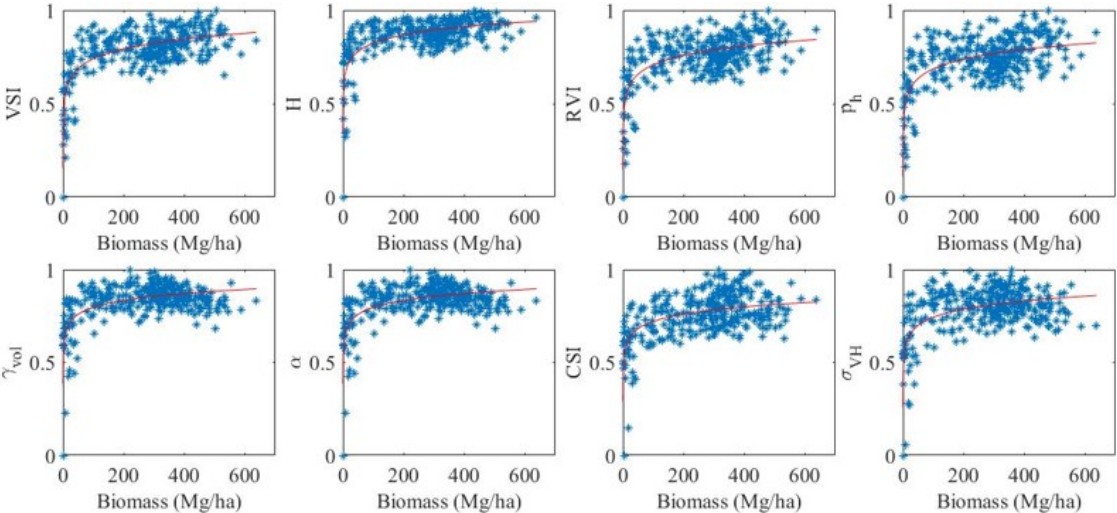

**Figure 8.** Model performance for the nonlinear regression of aboveground biomass (AGB) as function of polarimetric parameters.

It can be seen from Figure 8; VSI, H, RVI, $p_h$, and $\alpha$ are significantly correlated with referenced biomass measurements. Similarly, CSI and $\sigma_{VH}$ are moderately correlated with referenced biomass measurements. Similar consistent results were found for other PolSAR acquisitions (see Table 4). Similarly it can be seen from Table 4; regression results for 13 November 2018 acquisition are least correlated if compared to other acquisitions. This can be easily correlated with moist weather conditions during acquisition (see Figure 3). Among selected polarimetric parameters, VSI was found to be the most correlated with $R^2$ ranging from 0.45 (under moist conditions) to 0.62 (under dry conditions). Similarly, $\sigma_{VH}$ found the least correlated with $R^2$ ranging from 0.27 (under moist conditions) to 0.42 (under dry conditions). Regression results for 24 December 2018 acquisition are also moderately affected by moist conditions caused by rainfall few days before acquisition. The regression results for all selected variables are temporally consistent except of those affected by moist conditions caused by heavy precipitation.

As PolSAR-based AGB estimation at a higher level of forest biomass experiences saturation of PolSAR signal [5,28,29], in this study SAR signal saturation was observed mainly at stand level 300 Mg ha$^{-1}$. However, for few polarimetric parameters e.g., RVI, $p_h$, and CSI saturation was observed at stand level 400 Mg ha$^{-1}$.

### 3.3. Model Validation—Reference Biomass vs Observed Biomass

Based on regression analysis, models for VSI, H, RVI, $p_h$ and $\gamma_{vol}$ are selected for biomass mapping. To do so, these models were inverted and from resultant models, the AGB of study site was estimated.

To get a more realistic outcome, a low pass filter was applied to get relatively smooth AGB maps. Resultant biomass maps were validated using validation set (field based referenced biomass measurements). Validation results are shown in Figure 9. The resultant $R^2$ and *RMSE* are summarized in Table 5.

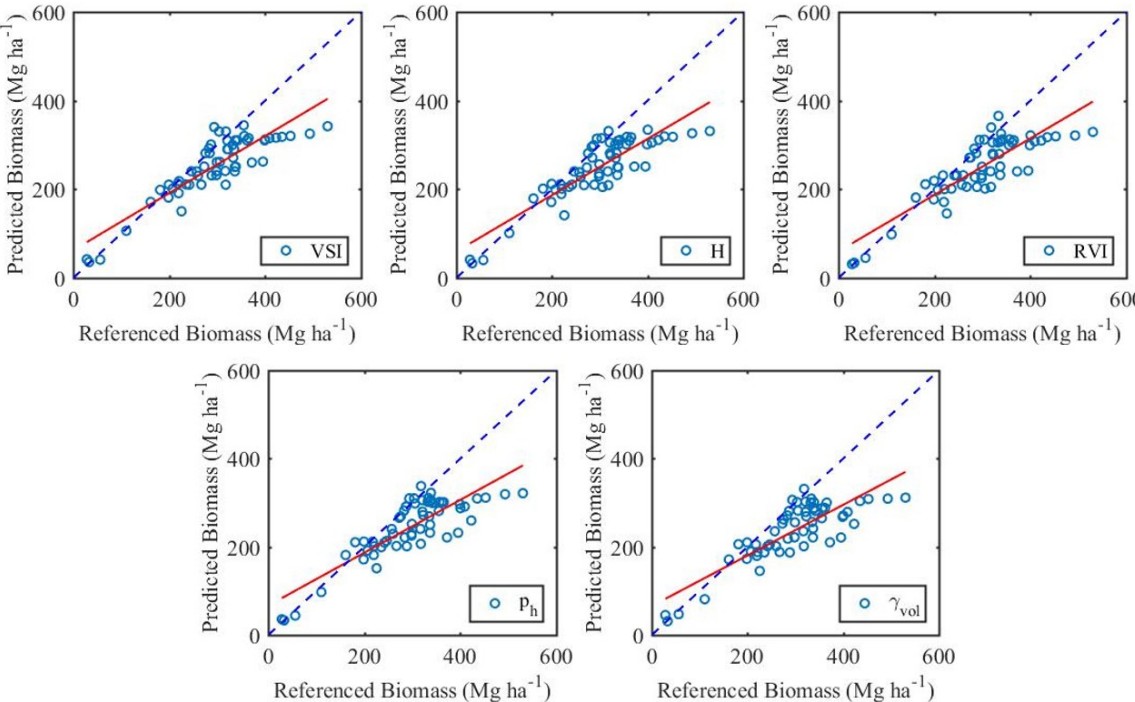

**Figure 9.** Scatter-plots of referenced biomass vs. predicted biomass from polarimetric parameters extracted from RadarSAT-2 scenes.

**Table 5.** Temporal AGB estimation with accuracy statistics.

|  | VSI | | H | | RVI | | $p_h$ | | $\gamma_{vol}$ | |
|---|---|---|---|---|---|---|---|---|---|---|
|  | $R^2$ | RMSE | $R^2$ | RMSE | $R^2$ | RMSE | $R^2$ | RMSE | $R^2$ | RMSE |
| **20 October 2018** | 0.76 | 34.43 | 0.73 | 36.88 | 0.71 | 41.88 | 0.70 | 39.56 | 0.68 | 39.87 |
| **13 November 2018** | 0.63 | 41.44 | 0.61 | 44.93 | 0.60 | 47.73 | 0.58 | 51.84 | 0.55 | 54.45 |
| **24 December 2018** | 0.71 | 34.76 | 0.67 | 38.33 | 0.66 | 44.11 | 0.64 | 44.43 | 0.58 | 46.53 |
| **31 December 2018** | 0.77 | 33.21 | 0.75 | 35.12 | 0.72 | 38.49 | 0.72 | 37.35 | 0.69 | 37.53 |
| **10 January 2019** | 0.73 | 34.88 | 0.71 | 36.98 | 0.70 | 42.42 | 0.66 | 40.43 | 0.66 | 41.23 |
| **17 January 2019** | 0.71 | 35.82 | 0.70 | 37.81 | 0.68 | 42.10 | 0.66 | 42.43 | 0.63 | 42.43 |

Validation results were found to be very promising with $R^2$ ranging from 0.77 (under dry conditions) to 0.63 (under moist conditions) and $RMSE$ ranging from 34.43 Mg ha$^{-1}$ to 35.82 Mg ha$^{-1}$ for VSI. Similarly, H, RVI, $p_h$ and $\gamma_{vol}$ validations results are also significant with $R^2$ ranging from 0.75–0.61, 0.72–0.60, 0.72–0.64, 0.69–0.55 and $RMSE$ ranging from 35.12–44.44 Mg ha$^{-1}$, 38.49–47.73 Mg ha$^{-1}$, 37.35–51.84 Mg ha$^{-1}$, 37.53–54.45 Mg ha$^{-1}$ respectively during dry and wet conditions. These results are consistent with other published research for forest AGB estimation [47,48]. However it can be seen from Figure 9, the estimated AGB is lower than referenced AGB. As C-band mostly interacts with leaves, main branches and under perfectly dry conditions can penetrate to the stem of tree, hence C-band is good to estimate AGB over low biomass regions.

### 3.4. Limitations

In this research, forest AGB was accurately estimated with the following limitations:

1.  Referenced biomass data collected through the field is not uniformly distributed throughout the study area. With more field data that are uniformly distributed throughput the study site cover major tree species can comprehensive understanding of true biomass conditions. However it is extremely difficult due to existence of wild-life.

2.  As region specific tree species allometric equations are not available for tree species in study site. Generic region specific allometric was used to calculate AGB using field data. Species specific allometric can give more accurate AGB estimates.

3.  As C-band is mostly sensitive to leaves and main branches, more precise AGB estimation can be done by developing synergy of polarimetric parameters extracted from C- and L-band PolSAR data.

## 4. Conclusions

This study presents tropical peatland forest biomass estimation using polarimetric parameters extracted from RadarSAT-2 images. Polarimetric parameters includes backscattering coefficient ($\sigma_o$), eign-decomposition parameters (H, $\alpha$), Yamaguchi decomposition parameters ($\gamma_{surf}, \gamma_{vol}$) and polarimetric indices (VSI, RVI, $p_h$, CSI) were used to modeled AGB. A detailed methodology for pro-processing of PolSAR images and AGB modeling is presented in this paper. Non-linear regression was used to model AGB as a function of polarimetric parameters. The regression result shows significant correlation between polarimetric parameters and referenced AGB. Selected regression models based on polarimetric parameters (VSI, H, RVI, $p_h$, $\gamma_{vol}$) were further used to generate biomass maps. Resultant biomass maps were validated with strong correlated was found between referenced AGB and predicted AGB with $R^2$ ranging from 0.77 to 0.58 ranging from 33.21 Mg ha$^{-1}$ to 37.53 under dry conditions, $R^2$ ranging from 0.63 to 0.55 and *RMSE* ranging from 41.44 to 54.45 under moist conditions.

PolSAR images acquired under perfectly dry conditions perform better than the ones acquired under moist conditions. A saturation point was observed at 300 Mg ha$^{-1}$ for VSI, H, $\gamma_{vol}$, $\alpha$ and $\sigma_{VH}$. However, the saturation point for models developed using RVI, $p_h$, CSI was observed at 400 Mg ha$^{-1}$. The selected models also shown temporally consistent behavior.

**Author Contributions:** M.M.W. designed, performed the experiment and wrote the manuscript. R.S. contributed significantly in field data collection and data processing. Y.J. revized the paper. J.T.S.S. supervised the whole research and provided suggestions related to research design and novel contributions. All authors have read and agreed to the published version of the manuscript.

**Funding:** RadarSAT-2 data were provided by Canadian Space Agency under project number SOAR-EI-5436. PCI Geomatica Software was provided by PCI.

**Acknowledgments:** Josaphat Laboratory (JMRSL) is supported in part by the 4th Japan Aerospace Exploration Agency (JAXA) ALOS Research Announcement under Grant 1024; the 6th JAXA ALOS Research Announcement under Grant 3170; Chiba University Strategic Priority Research Promotion Program FY2016-FY2018; SOAR-EI Canadian Space Agency (CSA) Project number 5436 FY2017–FY2019. JMRSL would like to acknowledge the support of PCI Geomatica for providing their software for this research.

**Conflicts of Interest:** The authors do not report any conflict of interest. The funding sponsors did not have any role in study design, data collection, analysis, data interpretation, manuscript writing and paper publication.

## Abbreviations

The following abbreviations are used in this manuscript:

| | |
|---|---|
| AGB | Aboveground Biomass |
| BMI | Biomass Index |
| CSI | Canopy Structure Index |
| DBH | Diameter at Breast Height |
| $h_p$ | Pedestal Height |
| PolSAR | Polarimetric SAR |
| RVI | Radar Vegetation Index |
| SAR | Synthetic Aperture Radar |
| VSI | Volume Scattering Index |

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
