# Peer review of "Tropical PeatLand Forest Biomass Estimation Using Polarimetric Parameters Extracted from RadarSAT-2 Images"

_land, doi:10.3390/land9060193_

Round 1

Reviewer 1 Report

The paper "Tropical Peat-land Forest biomass estimation using Polarimetric Parameters extracted from RadarSAT-2 images" is an original and great concept about the forest biomass in a tropical area. 

The paper meets the scientific criteria for making an scientific article and all the sections of the paper are realised in a profesional manner. 

The authors have described well the used methods and give sufficient details about. Also, the results completes the whole article.

The only suggestion for authors is to move lines 80-100 within the methodology section.

I am pleased that I had the opportunity to read and review this paper! 

Author Response

To:
Anonymous Reviewer,

Land MDPI Journal.

Subject: [Land] Manuscript ID: land-807422 - Response to Reviewer Suggestion

On behalf of all authors; I would like to thank the anonymous reviewer for sparing her/his time to reviewer this manuscript and providing valuable feedback. All authors agreed with the reviewer’s suggestion and the following suggestion provided by the reviewer is incorporated in the revised manuscript.

Suggestion from Reviewer: The only suggestion for authors is to move lines 80-100 within the methodology section.

Response from Author: Section 1.1 is moved under section 2.3 on page 5 line 125 and highlighted with green color in the revised manuscript.

Again, I would like to thank the reviewer for their valuable feedback.

Regards

Reviewer 2 Report

you have to merge the "Main PolSAR Concepts in the context of this study" with the method description
You need to simplify the method description
in line 48 change LandSAT with Landsat

Author Response

To:
Anonymous Reviewer,

Land MDPI Journal.

Subject: [Land] Manuscript ID: land-807422 - Response to Reviewer Suggestion

On behalf of all authors; I would like to thank the anonymous reviewer for sparing her/his time to reviewer this manuscript and providing valuable feedback. All authors agreed with the reviewer’s suggestion and the following suggestion provided by the reviewer is incorporated in the revised manuscript.

  1. Suggestion from Reviewer: You have to merge the "Main PolSAR Concepts in the context of this study" with the method description.

Response from Author: Section 1.1 is moved under section 2.3 on page 5 line 125 and highlighted with green color in the revised manuscript.

  1. Suggestion from Reviewer: You need to simplify the method description.

Response from Author:

Authors believed the detailed presented in method description is necessary to adequately present processing chain of PolSAR data. 

  1. Suggestion from Reviewer: in line 48 change LandSAT with Landsat.

Response from Author: In line 48; LandSAT is changed with Landsat and highlighted with green color in revised manuscript.

Again, I would like to thank the reviewer for their valuable feedback.

Regards

Reviewer 3 Report

I have structured my comments in two parts, first are observations about

the methodology, conclusions, and suggestions for further research, and

second are wording suggestions to begin to improve the writing quality, 

which needs significant work.

Part 1 

PolSAR Signal Saturation

Figure 8 is difficult to see what the precision is for the plots. Based on the curve and point distributions, it seems there are two problems. One is the saturation noted, where there is not much change in the parameters but the biomass changes. The other occurs on the lower part of the curve (near 0,0) where the parameters change rapidly, but the biomass much more slowly. The effect is that a given level of biomass can have more than multiple indices values associated with it.

It would be helpful to see these areas in more detail to understand the nature of the distribution. If the journal format permits it, maybe move this to an appendix, and feature one in more detail, since the plots are very similar. Another stategy might be to linearize data by transforming the
parameters.

Expand the discussion to explain what kind of forests correspond to the biomass levels. It would be insightful to understand in general the community properties that are related to the PolSAR scattering behavior. The discussion could be accomplished in broad categories say biomass 0 - 50
50 - 200 and greater than 200, or a division that identifies broad canopy groups relative to the PolSAR scattering behavior.

Scattering/Biomass/Forest Communities

The discussion about forest communities/scattering would also help guide additional work by identifying those areas that need more sampling. From the plots obviously most of the sites had greater than about 100 mg/ha, and few below that.

Part 2 Wording

The suggestions below should be a starting point, as significant modifications are needed.  In the wording suggestions (contained within parentheses), there are many examples of how to fix some consistent problems. The wording suggestions end on page 13.  Generally, here are some consistent problems;

  1. The use of semicolons  to separate thoughts within a sentence needs to be eliminated.  Use periods or restructure sentences.
  2. All references to specific figures and tables need to be capitalized (Figure 1 not figure 1)
  3. Watch out for passive sentence structure.  For example, "The forest canopy properties ..." rather than "The properties of forest canopies .."

Abstract

Are the polarimetric indices the PolSAR parameters referred to (PolSAR seems to be jargon
and the indices aren't defined as that). Need to define PolSAR before using.

20 space between Aboveground --> Above ground

28 "are home"

29 "However,"

32 "Southeast Asia"

33 "rainforests, especially in Kalimantan Island, are"

34 ",and forest burning, the stored C02 are returned to the atmosphere, and can alter atmospheric
composition, "

37 "Cost effective forest biomass assessment" The phrasing "Cost effective assessment of
forest biomass assessment" is passive writing as it is unnecessarily wordy.

44 Start new paragraph at "Application of optical remote sensing ..."

47 rephrase to "medium to high stand scale biomass"

48 misspelled Sentinal --> Sentinel

49 Start new paragraph at "As tropical regions .."

53 Passive wording change to "mapping and understanding ecological processes."

54 change to "forest structure depending on frequency"

55 X-Band scattering from forest areas mostly contains information about leaves and
small branches. C-Band scattering is governed largely by interaction with main
branches, while L-Band can penetrate **don't understand "until stem of the tree".
However, under ideal conditions P-Band can...."
66 "PolSAR signal, which is a well documented limitation of this technique."

67 New paragraph at "As natural forest ..."

68 "Few attempts have been made.."

71 "C-Band PolSAR data has not yet been performed."

72 "Furthermore,"

73 "polarimetric parameters were investigated for their multi-temporal AGB sensitivity."

74-79 "In section 2, fundamental PolSAR concepts are explained. Section 3 presents a description of study
area, PolSAR data, ancillary data, and biomass reference data. Also, a discussion of the PolSAR
data preprocessing and methodological framework are provided. Section 4 the central analysis
and discussion are detailed. Finally, Section 5 presents the conclusions and topics for further
research."

81 "by the scattering matrix displayed in equation 1 below:"

82 "polarization, respectively."

83 "needs to be applied"

86 "the covariance matrix is fundamental, as it characterizes the SAR image by its
corresponding scattering components,"

87 "Yamaguchi proposed a four component decomposition scheme [38], which decomposes
the coherency matrix into surface, double-bounce, and helix scattering."

Line Numbers Disappeared at Top of Page 4

Are there citations for BMI, CSI and VSI, should have a source for these or if the
authors are the source, they need to discuss the foundations of these indices.

Change "As BMI is not a ratio, and therefore is influenced by slope" to "As BMI is not a ratio, it
is relatively more sensitive to slope." NOTE: Even ratio indices can be sensitive to slope. The
well known NDVI has been shown to have slope influences, as the NIR component can be
reflected preferentially.

Change "As chosen study area ..." to "As the chosen study area ..."

Change "High values of result ..." to "High values result .."

Line under Eqn. 12 should that HH = HH be HH = VV ? the VV term was dropped in Eqn. 13

121 "from the Canadian Space Agency ..."

127 "... precipitation measured from the nearest meteorological ..."

140 Generally the use of "shows" is discouraged, I would switch to displays,depicts,
illustrates, presents, exhibits,

149 "Eign"

Table 3 "Canopy"

"If X is the independent variable and Y is the dependent variable, then the logarithmic
regression equation can be written as;"

169 "If the relationship between dependent and independent variables is perfectly
linear then ...."

173 "summarized in Table 4."

173 "In order to get AGB as a function of polarimetric parameters, the
resulting regression models were inverted."

174 "Biomass maps were generated for all acquisitions using the inverted
regression equations, and then validated with the previously described
validation maps."

178 "First we discuss the temporal consistency, and interdependency between polarimetric parameters,
and their suitability for AGB estimation."

179 "Then we describe the regression results between polarimetric parameters, and reference
biomass measurements. Finally, the validation results are described in detail."

183 "As PolSAR images were acquired under different weather conditions (see Figure 3), its
important to analyze the impact of the different weather conditions at the time of
acquisition."

184 "To examine the weather influence, the PolSAR parameters were extracted from temporal
images, and evaluated to assess their temporal consistency, interdependence, and suitability
for forest above ground biomass estimation across rainy and dry conditions.

183 "(Figure 3)"

191 "(listed in Table 3)"

193 "(listed in Table 3)"

197 "listed in Table 4. All polarimetric parameters were normalized
between 0 and 1, see Figure 8, enabling comparison of the polarimetric parameters,
which have different dynamic ranges."
200 "Figure 8 depicts the regression of PolSAR measurement and indices with biomass.
VSI, H, RVI, ph, and alpha are significantly correlated with biomass, whil CSI
and sigma VH are moderately correlated. Similar consistent results were found
for other PolSAR acquisitions (see Table4). The least correlated data set is the
output from November 13, 2018 the PolSAR for which was acquired recently after a
precipitation event."

205 "Considering all PolSAR acquisitions, VSI was the most consistently correlated with
R2 ranging from 0.45, under moist conditions, to 0.62, under dry conditions. By
contrast sigma VH was found to be the least correlated with R2 ranging from 0.27,
under moist conditions, to 0.42, under dry conditions. Regression results for
December 24, 2018 were moderately affected by moist conditions from rain a few days
before acquisition. Overall, temporal consistency for all measures were affected by
rainfall closely predating PolSAR acquisition."

212 "In this study SAR signal saturation was observed at the stand level at approximately 300 Mgha-1,
which is consistent with previous research [27-29]."